# Intentional Observational Clinical Research Design: Innovative Design for Complex Clinical Research Using Advanced Technology

**DOI:** 10.3390/ijerph182111184

**Published:** 2021-10-25

**Authors:** Tetsuya Tanioka, Rozzano C. Locsin, Feni Betriana, Yoshihiro Kai, Kyoko Osaka, Elizabeth Baua, Savina Schoenhofer

**Affiliations:** 1Department of Nursing Outcome Management, Institute of Biomedical Sciences, Tokushima University, Tokushima 770-8509, Japan; 2Institute of Biomedical Sciences, Tokushima University, Tokushima 770-8509, Japan; locsin@tokushima-u.ac.jp; 3Christine E. Lynn College of Nursing, Florida Atlantic University, Boca Raton, FL 33431, USA; 4Graduate School of Health Sciences, Tokushima University, Tokushima 770-8503, Japan; fenibetriana@gmail.com; 5Department of Mechanical Engineering, Tokai University, Hiratsuka 259-1292, Japan; kai@keyaki.cc.u-tokai.ac.jp; 6Department of Nursing, Nursing Course of Kochi Medical School, Kochi University, Kochi 783-8505, Japan; osaka@kochi-u.ac.jp; 7Graduate School, St. Paul University Philippines, Tuguegarao 3500, Philippines; ebaua@spup.edu.ph; 8Anne Boykin Institute, Florida Atlantic University, Boca Raton, FL 33431, USA; savibus@gmail.com

**Keywords:** intentional observation, clinical research, novel research design, nursing, advanced technologies, healthcare, mixed methods research

## Abstract

The growing use of robots in nursing and healthcare facilities has prompted increasing research on human–robot interactions. However, specific research designs that can guide researchers to conduct rigorous investigations on human–robot interactions are limited. This paper aims to discuss the development and application of a new research design—the Intentional Observational Clinical Research Design (IOCRD). Data sources to develop the IOCRD were derived from surveyed literature of the past decade, focusing on clinical nursing research and theories relating robotics to nursing and healthcare practice. The distinction between IOCRD and other research design is the simultaneous data generation collected using advanced technological devices, for example, the wireless Bonaly-light electrocardiogram (ECG) to track heart rate variability of research subjects, robot application programs on the iPad mini to control robot speech and gestures, and Natural Language Processing programs. Even though IOCRD was developed for human–robot research, there remain vast opportunities for its use in nursing practice and healthcare. With the unique feature of simultaneous data generation and analysis, an interdisciplinary collaborative research team is strongly suggested. The IOCRD is expected to contribute guidance for researchers in conducting clinical research related to robotics in nursing and healthcare.

## 1. Introduction

Particularly in a disciplinary practice profession such as Nursing, it is critical and appropriate to deliberate on the complexities of phenomena of interest in its science that demand judicious understanding [1,2]. In past decades, the popularity of mixed method designs has increased strategically [3], and researchers have deemed it more practical to apply these designs in rigorous inquiries involving complex phenomena [4].

One of the indications of mixed method designs is the research question [5]. Having complex questions requires multiple ways of evidencing for testing hypotheses or describing phenomena under study. Mixed methods research, as a combination of two or more types of research methods/designs, is advantageous toward finding discipline-related knowledge that can inform the ontology, epistemology, and axiology of the discipline [6]. With data generation and analysis increasingly directed to uncover findings that can inform the discipline, mixed methods research has become a highly relevant approach in current healthcare research [7].

The basic requirement regarding the credibility of research findings in mixed-method research is to sustain the demands of rigorous data generation, analysis coordination and interpretation of data. While the complexity involved in mixed methods designs can be viewed as a disadvantage [8], these methods are also found useful in gaining insights into clinical nursing research problems [9]. Palinkas et al. [10] explained that conducting mixed methods research poses several challenges from design processes to analysis and to dissemination. However, the desire for quality outcomes of clinical research in the absence of mixed methods research results in multiple but separate publications of either quantitative or qualitative findings, rather than with integrated findings from a mixed methods research protocol.

Advancing technologies such as healthcare robots and sophisticated technological devices guiding protocols for studying complex clinical nursing research phenomena provided the impetus for developing the new research design. This approach must contribute an innovative research design predominantly intended to generate data using advanced technological devices in clinical research, contributing to nursing science [11].

This paper aims to discuss the development and application of the Intentional Observational Clinical Research Design, featuring the use of advanced technological processes and measurements in complex clinical nursing research.

## 2. Materials and Methods

A descriptive approach was conducted by reviewing and analyzing literature systematically focused on technological devices and theoretical frameworks in nursing and healthcare and to develop novel mixed methods research.

Data sources included surveyed literature during the past decade regarding mixed method designs for complex clinical nursing research approaches. Inspired and supported by twenty years of experience, Tanioka and co-researchers brought meaning and focus to clinical nursing research methods, furthering the development of nursing science. The development of innovative research designs that match these advances is required. Based on their clinical research experiences and supported by theoretical underpinnings and surveyed literature data, the Intentional Observational Clinical Research Design (IOCRD) evolved as a novel clinical nursing research design.

### 2.1. Review of Literature

With the multiplicity of healthcare phenomena in highly technological environments, and significant developments in nursing and health sciences [12,13], the impetus to develop novel methods and designs of research is recognized. Allowing the generation of data that meet rigorous standards demanded in developing knowledge, particularly in nursing science, new designs should be focused on generating and collecting data that answer complex research questions. These needs involve advancing healthcare technologies, particularly those with healthcare robots, human nurses, and the intermediary roles of healthcare professionals.

Studying the phenomenon of transactive patterning with intelligent humanoid robots and human caring interaction processes requires combinations of research methods. In particular, simultaneous data collection and generation must measure quantifiable actions, while describing and explaining communication patterns of human expressions such as compassion, involving transactive relationship engagements with robots, patients, nurses and their intermediaries. Nurse researchers are expected to intentionally observe the transactions among healthcare robots, humans and their intermediaries to obtain credible, appropriate and accurate research data.

With technological advances lending sophistication in the generation and collection of quantitative and qualitative data, flexibility and utility can be expected in several ways [14,15,16,17,18]. It is useful to combine several research methodologies to clarify highly complex phenomena. However, a research approach that recognizes advanced technology as integral to the simultaneous data collection and generation has not yet been duly explored.

To describe the distinction between IOCRD and other research designs used in human–robot research, particularly a mixed-method design, a review of the literature was conducted systematically. Relevant articles were retrieved if they met the inclusion criteria: (1) publication date between 2011 and 2021, (2) articles in English, (3) were fully accessed, (4) applying/discussing mixed-method design in the context of social robot and human research. To retrieve broader studies, theoretical papers, discussion papers, and review articles that discussed mixed-method and human–robot interaction were also included. Articles were excluded if the main findings did not discuss social robot and human interaction and did not utilize or discuss a mixed-method design. The search process was conducted using electronic databases, including PubMed, ScienceDirect, and SCOPUS, using the combination of different keywords by Boolean strategies (AND, OR). The keywords included “mixed-method,” “robot,” “social robot,” “hospital,” “clinic,” “laboratory research,” and “simultaneous data collection”. Additional articles were also searched by checking the reference list of the relevant studies.

Findings show that the distinction between IOCRD and mixed method is centered in two points: simultaneous data generation and the utilization of advanced technologies.

A mixed-method study enables researchers to combine qualitative and quantitative data collection and analysis in a single study [19]. To conduct a mixed-method study, researchers can collect quantitative data through administering an instrument to research subjects and then analyze the data. Afterward, researchers can select outlier cases and conduct interviews with individuals from the sample for collecting qualitative data. Alternatively, researchers may want to perform an observation first in the research setting to collect qualitative data followed by surveying individuals in that setting to collect quantitative data and combine the findings of both data [19].

In review articles, studies applying mixed method design collected quantitative and qualitative data separately, mostly by collecting quantitative data first through asking participants to fill out the questionnaire, followed by interview or focus group discussion [20,21,22]. Before data collection, several interventions were conducted, such as showing the video of older people living with a robot to research subjects [23], allowing participants to directly interact with the robot [20,21,24]. A study by Hebesberger et al. [25] conducted an observation of participants’ interaction with the robot between the interview and delivering an online questionnaire regarding the acceptance of robots in care hospitals. Meanwhile, Wiart et al. [26] compared several interventions involving robotic gait training by which these interventions were followed by assessment (quantitative data) and qualitative interview. Most of the studies in social Human–Robot Interaction (HRI) focus on the role of the robot as interactants rather than as assistants to human–human social interactions. HRI studies of robots intervening in human–human interactions vary widely in their scope, and are scattered across domains of application, using very different robot designs in various contexts. However, some of these studies were conducted in lab settings while others were in more naturalistic settings such as nursing homes [27].

In IOCRD, observation, quantitative and qualitative data were collected simultaneously in the clinical setting [28].

For example, while participants were interacting with robot, the interaction was an audio-video recorded with digital video cameras (observation). During the interaction, participants wore wireless Bonaly-light electrocardiogram (ECG) equipment to measure their heart rate variability (HRV) for quantitative data so that the changes in their autonomous nervous activity during interaction with the robot can be tracked in real-time. Additionally, their conversations with robots and with intermediaries or with researchers during and after the interaction (qualitative data) were also audio-video recorded. After the collection of these data, all data were analyzed.

For example, if in one scene, the participant was observed to laugh with the robot (observation data), at the time the scene occurred, the participant’s HRV was checked and analyzed to examine if there was change in the sympathetic or parasympathetic nervous activity before, during, and after that scene (quantitative data). Moreover, the conversation was transcribed, and the researchers analyzed the transcriptions of the recorded conversation and focused more on that particular scene (qualitative data).

To analyze human–robot interactions, it is essential to synchronize all the data to be analyzed. For example, a synchronizing signal can be used to synchronize all the data or using a radio clock to fit all the data reference times, which are continuously verified.

The second distinction between IOCRD and mixed method is the utilization of advanced technologies for data collection and analysis.

In the reviewed studies, quantitative data were collected through questionnaires [20,23] while qualitative data were collected by semi-structured interviews [23] or focus groups [20]. One study reported a modification of the setting before data collection by installing the robot, networking, video cameras, a microphone, and map of the patient’s bedroom space [21] and monitoring patient’s interaction with the robot; however, no further information was given on whether the monitored interaction was analyzed and how to analyze it. Instead, it was informed that after the interaction, participants completed questionnaires and were interviewed regarding their perception and acceptance of the robot.

In IOCRD, data collection and analysis have utilized advanced technologies, for example, digital video cameras, wireless ECG, and natural language processing programs.

### 2.2. Theoretical Underpinnings of the IOCRD

Clinical practice settings are continuously undergoing major changes that impact the delivery of quality healthcare. In situations such as this, the application of human-to-human and human-to-intelligent machine practice processes require dependable processes of care. To ensure that this demand for quality is met, improvements of practice protocols consistent with safe, secure, and dependable protocols are developed through rigorous research. The IOCRD was designed to assuredly meet this necessity. Two theories of nursing provided the theoretical underpinnings supporting the development of the IOCRD. These are the Transactive Relationship Theory of Nursing [29] and the Model for the Intermediary Role of Nurses in Transactive Relationships with Healthcare Robots [30].

#### 2.2.1. Transactive Relationship Theory of Nursing

Tanioka [29] declared that studying transactive engagement phenomena derived from the Transactive Relationship Theory of Nursing (TRETON) does not readily respond to traditional mixed method designs, instigating the impetus to develop an innovative research design. Desired characteristics of the IOCRD required that it be capable of retrieving precise, reliable and significant clinical research outcome data.

The five assumptions of TRETON are identified and described, illuminating the ways that the IOCRD was developed, rationalizing its application.
(1)Nursing is a relationship between human beings (human persons) and intelligent machines (healthcare robots).*Artificial Intelligence (AI) influences the healthcare robot’s functionality as an intelligent machine. With its capabilities as a human caring entity, it is necessary for healthcare robots to express caring in nursing to enhance the quality of nursing care through “delegated” healthcare tasks.*(2)Nurses use technologies of care for their practice.*Nurses are adept at technologies that aid their practice of nursing. Nurses must always exercise sensibility with AI-programmed machines, as transactional relationships are guided by the ethical use of AI in nursing.*(3)Intelligent machines with AI can mimic human-to-human interactions.*Various AI levels provide “intelligent” versions of the humanoid robot capability of interactive discourse and seamless physico-mechanical movements. The AI of healthcare robots gives them the ability to engage in functional relationships, to possess high-level AI programming and to extend the interactive discourse with human beings.*(4)Human-to-intelligent machine relationships are technology dependent.*From mobility to physical capabilities, robots depend on sophisticated technologies, from enhanced robotic performance to “super-intelligent” discursive abilities.*(5)Transactive relationships are guided by ethics in nursing.*Regardless of the level of sophistication in physico-mechanical functions and intelligence, healthcare robots are expected to have a programmed response system guided by ethico-moral sensibilities.*

#### 2.2.2. Model for the Intermediary Role of Nurses in Transactive Relationships with Healthcare Robots

Healthcare robots are intelligent machines used in healthcare settings [31]. However, their functions and performance are still continually being developed [32]. The complexity of nursing phenomena is compounded with the introduction of healthcare robots. One of the primary goals in the development of the IOCRD was the need to address this complexity effectively in nursing research processes to meet the demands of future practice. Osaka’s Model for the Intermediary Role of nurses in Transactive relationships with Healthcare robots (MIRTH) [30] support the intricacy of data collection and generation in complex clinical research situations with human–robot transactive relationships.

The MIRTH has five assumptions:
(1)Humanoid robot performance requires an intermediary for their effective and safe use [30,33].*The importance of the functions of intermediaries in robot-human situations is heightened as the intermediary is inextricably linked with the patient and HR;*(2)Robots are used for rehabilitation, recreation, and caring for older adults [34].*This assumption describes the variety of functions of robots specifically for older persons;*(3)High-quality care with robot–human relationships is guided by ethical and moral standards of nursing [35].*With human beings as patients, healthcare robots are integral to human health care. This relationship must be linked with considerations of beneficial effects founded on justice and goodness;*(4)Technologies of health and nursing are elements for caring [36].*The utility of advancing technologies founded on competency as expression of caring provides opportunities for innovating human caring practices;*(5)Nursing is both a discipline and a profession [37].*It is the responsibility of nurses as professionals to practice nursing grounded in discipline-related knowledge of nursing. The most important attribute in nursing is the relationship expressed as caring between patients and nurses as the interactive engagement that gives meaning to living their own lives.*

## 3. Discussion

This article discusses the development, utilization, and application of the Intentional Observation Clinical Research Design (IOCRD), a research design using advanced technological processes and measurements in clinical nursing research. Four questions were posited to describe and explain the design: (1) What is the Intentional Observational Clinical Research Design? (2) How is the IOCRD used in Research? (3) What are the requisites of the IOCRD? (4) How can the IOCRD be applied in clinical nursing research?

### 3.1. What Is the IOCRD?

The IOCRD is a composite of observational research approaches designed to generate rich data from a simultaneous procedure to answer complex questions relevant to phenomena of concern. While mixed methods research designs have been widespread, the IOCRD as a unique research method was envisioned to integrate several simultaneously measured variables/data to respond to the complexity of researching phenomena contributing to nursing science. The IOCRD as an emergent methodological approach will increase the credibility of nursing research, thereby supporting the significance of human-healthcare robot relationships.

In developing the IOCRD, the following requisite processes were considered: (a) Evidentiary Data: a survey of the current literature regarding methodological designs for conducting clinical nursing research involving complex phenomena, (b) analysis: determine the integration of literature on intentional observations in collecting/generating clinical research data. (c) Findings: creating the framework of the Intentional Observational Clinical Research Design; (d) development: to develop the IOCRD; (e) usage and validation: to validate the IOCRD, studies were described on the basis of the procedure for data collection and analysis using the IOCRD; and (f) continuous improvement: systematic discoveries and feedback loops of continuous improvements of the processes involving the IOCRD.

The IOCRD was designed with complex methods combining phenomenological and experimental data gathering procedures. It is critical to clarify the accurate and appropriate conclusions regarding the research questions about the phenomenon of interest particularly grounded in the TRETON [29].

Table 1 shows the differences between the IOCRD and other research designs. The approach of the IOCRD is a quantitative, qualitative, and intentional observation. The aim of the IOCRD is to address the transactive engagement phenomena of human-intermediaries-robot and advanced technologies. Moreover, a unique innovative feature of the IOCRD is intentional observation using advanced technologies to understand complex relationships among these phenomena.

Figure 1 shows the concurrency between three data collection/generation modes of the IOCRD—the quantitative, qualitative, and mixed methods. In the IOCRD, all three forms of data collection and generation occur simultaneously. Therefore, in the data analysis phase, researchers have all data funneled to the spout. With all three sources of data analysis represented by the funnel’s mouth and analysis and interpretation at the base of the funnel, all data are being interpreted simultaneously vis-a-vis each other, with findings integrated at the spout section of the funnel. The IOCRD is the output of the integration of intentional observations to collect/generate data. This output produces innovations.

### 3.2. How Is the IOCRD Used in Research?

A truly organized and deliberate interdisciplinary group comprises the team through which the IOCRD can operate. Through this team, each member researcher’s expertise contributes to the knowledge and skill required to generate, analyze, and interpret data using the IOCRD to answer interdisciplinary complex research questions that require interdisciplinary answers. Furthermore, intentional observations in clinical research allow young researchers as novices in research to render intentional observations in conducting quantitative experimental research studies while being guided/supervised by experts of specific technological devices that are used to gather data. It appears that experts can give clearer guidance with intuitive specificity based on experiencing research situations. With years of experience, intentional research data collection and generation with the utility of intentional observational design and focus can be achieved with considerable confidence.

Similarly, these researchers can describe how interdisciplinary research collaborations (especially nursing and engineering and other disciplines involving technologies) can comprise a viable procedure for data collection and generation. Importantly, it is educationally essential for researchers who are not accustomed to experimental methods to make in-depth experiment preparations and engage in analysis together with these expert researchers. Furthermore, since it is necessary to perform a participant observation in parallel with these experiments, it is impossible for one researcher alone to perform experiments efficiently.

Today, robots related to various nursing tasks (excretion support, mobility support, communication, etc.) are being developed in an increasing number of countries worldwide [38]. Thus, it can be expected that an increasing number of technologies will enter the nursing setting. Nurses should take an active interest in these technologies and reaffirm their nursing expertise. For example, assuming that a new communication robot for supporting older people has been marketed, the description of performances of a communication robot should be observed intentionally to understand its contents. It is also essential to consider what problems occur when using it to support older adults and whether it will improve nursing. The IOCRD should be used as a framework for determining needed researcher expertise. Therefore, it is essential to ask, “What is the analytical/synthetical method designed to achieve the research purpose(s) that was made rigorous through complex data of both quantitative and qualitative forms?” The IOCRD is used in these situations.

### 3.3. Requirements for Using the IOCRD?

Researchers should have a positive interest in combining new technologies and using these interests for nursing research. In a survey conducted around ten years ago, nursing scholars and nurses were negative or indifferent to robots (technology) performing nursing tasks [39,40,41,42]. If a nurse has no interest in technology [36], this position suggests that such a non-interest leads to a dangerous situation involving poor nursing care. However, if nurses have an active interest in technology and appreciate its potential use in nursing and healthcare, it would contribute substantially to nursing knowledge development.

A critical evaluation of new technology is essential for protecting patient safety and advocating human rights. It is also essential that healthcare field researchers know that engineers are thinking of practical applications and evaluating tests toward innovative apparatuses that are narrow gateways to technological development.

It is argued that developments of knowledge derived from nursing theories and rigorous research demand data-generating technologies that foster competencies with various sophisticated research approaches determined as intentional observational processes. These five points are the requirements of the IOCRD. (a) Required Researcher Qualifications to Use the IOCRD: It is critical for each researcher to have an exact role and conduct deliberate observations onsite under an interdisciplinary research team’s leadership. (b) Training of Researchers within the Process of Interdisciplinary Collaborative Research: The outcome of research studies influences the quality of care of persons. In interdisciplinary collaborative research, the researcher needs to be imbued with “compassion for others” as part of human nature. Observers need to be trained in the phenomenon to be observed. Furthermore, observers then compare notes, learning what observers from another field will “see” that others did not “see.” (c) The Required Proficiency for Researchers Using the IOCRD: Competency to improve observation and insight about phenomena to be studied and to communicate with collaborators. “Observation” is the ability to perceive accessible data or information that is of interest. (d) Strive to Speak in a Common Language: If researchers from one discipline explain something using a technical term from that discipline, researchers from another discipline will not understand it. (e) Has an Appropriate Interest in Technology: Researchers should have a positive interest in compassion for others, combine new technologies and use these interests for nursing research to contribute to improving the quality of nursing and healthcare.

### 3.4. How Can the IOCRD Be Applied in Practice?

#### 3.4.1. Applications of the IOCRD

The analysis process for data derived from using IOCRD involves two known research approaches since this design involves mixed methods. It starts with quantitative analysis and is followed by qualitative analysis. In the IOCRD, researchers who excel in experimental methods, multivariate analysis methods, and qualitative research designs can gather interdisciplinary research data through clinical experiments. It is emphasized in the IOCRD that researchers intentionally observe overall relationships according to the study’s purposes. Furthermore, each researcher plays an exact role and function to conduct deliberate observations designed and supervised by an interdisciplinary research team. Considering the multiplicity of healthcare problems of a highly technological nature and the emergence of significant nursing science developments, it will provide an impetus to develop research methods that allow the generation of data that meet rigorous standards as demanded in developing knowledge, particularly in nursing research. The following examples from clinical research illustrate how information gathered during one data collection can generate different types of data.

To understand the phenomenon and appreciate the research and development issues, a required description of the characteristics of the research process is essential, and careful planning of the procedure for data collection must be designed. The framework of the IOCRD is described and explained.

The IOCRD can be effective in measuring both quantitative and qualitative data. For example, in research studies requiring physiological data and relational interactive data between healthcare robots and older adults, (1) HRV of older adults can be evaluated using an electrocardiogram, and (2) transactive relational data for healthcare robots, older adults, and intermediaries can be recorded by digital video recording. The HRV analysis can be performed by synchronizing the time between the digital video camera and the ECG monitor, and results of participant intentional observation using video camera data and HRV analysis can be analyzed as well [28,43]. To realize the simultaneous acquisition of quantitative and qualitative data, it is necessary to synchronize all measurement time lines of the experimental (quantitative) and observational (qualitative) study.

Figure 2 shows an example of the synchronized image of (1) HRV evaluated using ECG, and (2) transactive relational data recorded by digital video recording, and (3) participant Intentional observational data.

#### 3.4.2. Intentional Observation 

Figure 2 also shows the clinical experiment on optimization of human–robot interaction in improving communication between a humanoid robot and older adults through dialogues. To observe relationships between robot and older adults in clinical practice, first, relationships between the talking robot and older adults are intentionally observed.

Intentional observation calls for descriptions of what the observers were “seeing” vis-a-vis the robot, the older adult and the intermediary. Furthermore, the interactions at that time are captured in digital video [44,45]. This case clarified points for fine-tuning of the current function of the humanoid robot’s dialog pattern to improve optimal communication between a humanoid robot and older adults. The data were collected by intentionally observing and recording dialogues between older adults and a Pepper-robot using video cameras. Data from transcriptions of conversation captured from video and field notes were analyzed, focusing on human–robot interaction. In addition, real-time autonomic nerve activity analysis shows the sympathetic–parasympathetic balance. In the change of NU (ratio of sympathetic and parasympathetic nervous activity), researchers can confirm this conversation scene shows that the patient’s sympathetic nerve activity is dominant.

#### 3.4.3. Qualitative Data Generation

Figure 3 shows the effectiveness of healthcare robots and the intermediaries’ role [30,46] between healthcare robots and older adults.

The natural language processing (NLP) program, enabling the robot to converse with humans, excluded a wait time for the interval in order for older adults to respond, causing a difficulty for the robot to respond accurately. Therefore, there were some concerns indicating failure to communicate well. It is essential to consider the intermediary role connecting the robot and older adults and their role, even if the robot is not sophisticated enough to be useful for rehabilitation and dialog with older adults [44]. Participation observation and interviews are essential to clarify this role. The experimental method responds to the cause-and-effect relationship as the focus of studying the phenomenon of interest. However, as it is generally the case with phenomenological approaches, descriptions and explanations of phenomena of concern are derived from experiences in which language is critical. For the novel innovation of intentional observation, participant observation is critical during a clinical experiment, particularly in phenomenological research.

The phenomenon of interest in this study was the effectiveness of healthcare robots and the role of intermediaries. Data were generated by following the procedure of data generation, namely: (a) Clarifying the phenomenon of interest—in this study, the phenomenon of interest was the transaction between healthcare robots and patients with nurses as intermediaries; (b) to determine the data required to be able to confirm the significance of the qualitative data, instruments were used such as digital video, NLP data log from the robot; (c) intermediaries create relationships with the robot so that they promote the health and safety of older adults and increase their enjoyment through physical and socialization activities. After the clinical experiment, it is critical to interview the intermediary about his/her-specific role behaviors in relation to the older adult during the engagement; and (d), analyzing the data—both qualitative and quantitative data can be generated following an established protocol for data gathering for both qualitative and quantitative data methods.

The dialogue was difficult without an intermediary role because it was difficult for older adults to understand the speech rate and tone of the robot. However, even though healthcare robots’ communication was insufficient in terms of timing and content, older adults tried to communicate with care robots. The intermediary’s role was indispensable to make up for an inadequate robot in an older adult interaction.

### 3.5. Uniqueness of the IOCRD: Simultaneous Data Collection and Generation through High-Tech Devices

Figure 4 and Figure 5 illuminate research studies that employed the IOCRD, demonstrating its usage in clinical research. In this research design, researchers can continuously and simultaneously observe and understand the physical data among various data acquired, which show physical changes of older persons (e.g., HRV and range of motion data) during interactions between robots and humans from moment to moment.

Researchers can also examine “contents of motivational expression from robots to older people (Figure 4),” and “effects of rehabilitation for improving shoulder joint range of motion of older people (Figure 5).” Innovation is derived from these comprehensive considerations.

Figure 4 shows the scene of range of motion of the shoulder exercises instructed by a Pepper Robot.

Figure 5 shows a typical example of the synchronized image of (1) HRV evaluated using ECG and (2) transactive relational data recorded by a digital video recording during the range-of-motion exercise.

## 4. Implications for Nursing Science and Healthcare Using Advanced Technologies

### 4.1. Nursing Research

In studying nursing as a discipline and a profession, one thing is certain, that advancing nursing science and the development of nursing practice grounded in perspectives that are of nursing rather than other sciences is essential. Researchers of nursing consider focusing on studying nursing phenomena grounded in theories of nursing. Transactive engagements such as nursing practice encounters between persons, healthcare robots, and nurses as intermediaries [29] highlight the contribution of the theory of Technological Competency as Caring in Nursing [36] in revealing nursing actions in situations such as these and informing the quality of nursing care practice.

### 4.2. Nursing Education

Nursing education relies on nursing research for a sound scientific basis for nursing practice, and nurse researchers are prepared for various research roles throughout the curriculum. Thus, the nursing curriculum and its content must be advanced with the appreciation of nursing as a caring science, grounded in human philosophical perspectives, incorporating innovative and nursing-relevant research such as IOCRD, conceptualized within explicit nursing theoretical frameworks. IOCRD is an excellent example of the type of caring science advancement advocated by the document, “Guiding Principles for Transforming Curriculum through Integration of Technology as Expression of Caring” [47].

### 4.3. Nursing Practice

Nurses in practice settings play a key role in serving as intermediaries in intervening during dialogues between older adults and robots. Clinical nurses also contribute to the conduct of research. With focused training, the role of the observer as described in this paper is well within the capability of nurses caring for older adults, and in fact, the clinical nurse brings important first-hand awareness to this role.

### 4.4. Recommendations for Interdisciplinary Collaboration for Healthcare

From the viewpoint of NLP, it is vital to observe a robot’s verbal and nonverbal expressions to improve linguistic expressions. As for the expression of words, one should intentionally observe whether robots express natural words to persons from the viewpoint of nursing care. Engineers can create an interaction database for normal NLP. Therefore, nursing researchers have much to contribute to creating a caring NLP database for persons such as older adults. It is also essential to inform NLP researchers in terms of the appropriateness of expressions declared as caring expressions. This idea is more evidence of a revolutionary thought in data collection and generation.

Improving communication between healthcare robots and human persons and optimizing a structured dialogue can enhance transactive engagements and participation. It is necessary to create a Caring Dialogue Database for healthcare robots to know the patient/client and share aesthetic experiences of human–robot interactions. It is crucial to develop a dialogical pattern that allows the healthcare robot to sympathize with human beings, particularly older adults. To advance the nursing innovation, the participation of nursing researchers as described above advances the technology of caring and advances the form of nursing in cooperation with healthcare robots and, simultaneously, machine learning for NLP. In this way, nurse researchers can contribute to the development of AI for healthcare.

### 4.5. Limitations

Limitations are identified when applying the IOCRD. These include the availability of resources and/or advanced technologies for data collection and analysis, and the risk of bias during observations. Therefore, confirming the resource availability in research settings and good collaboration among members of the research team are highly suggested in applying IOCRD. In order for IOCRD to be a standardized model and/or a guideline model for future research, its utility needs to be validated in various areas such as in nursing, engineering, sociology, psychology, and medicine where real-time data, which is simultaneous data generation, are collected/measured using advanced technological devices. Comparing the results obtained by other designs with the results acquired by IOCRD is an issue that needs clarification and can be performed in the future.

## 5. Conclusions

With advancements in healthcare robotics, the advent of human–robot interactive phenomena continues to challenge the rigors of clinical nursing research methodologies. The IOCRD is a breakthrough research innovative design focusing on technological advances in measuring complex clinical nursing research phenomena. The IOCRD as a novel research design is composed of observational research approaches designed to generate rich data from a simultaneous procedure involving highly technological instruments and tools to answer complex questions relevant to phenomena of concern. As a novel research design, its methodological possibilities encompass data generation using advanced technologies. It can be used in deriving data regarding complex clinical research phenomena from nursing situations. With advancing technologies impacting clinical nursing research and in practice settings, the IOCRD allows for the simultaneous data generation and analysis procedure, especially those that involve intelligent healthcare machines such as humanoid healthcare robots in studying nurse transactive engagements.

## Figures and Tables

**Figure 1 ijerph-18-11184-f001:**
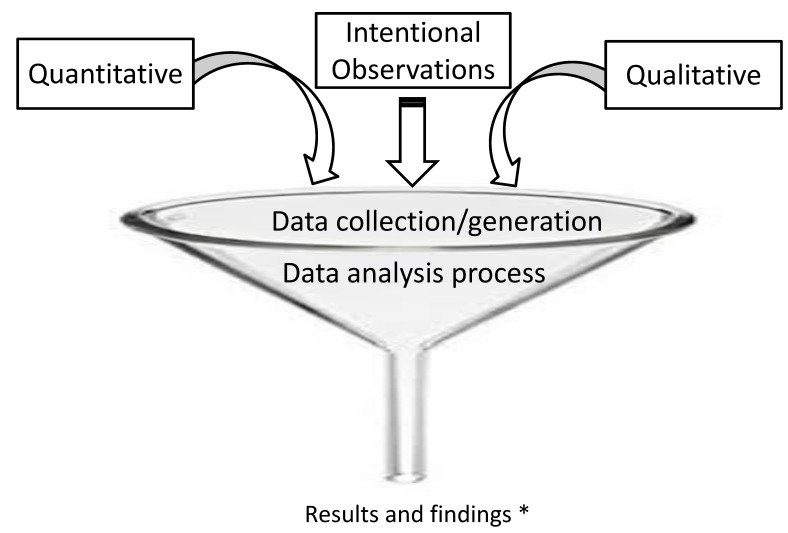
Illustrating the concurrency between three data collection/generation modes of the IOCRD. * Integrated findings are coming out from the spout of the funnel. The IOCRD is the output of integration of intentional observation to collect/generate data. This output produces the innovation.

**Figure 2 ijerph-18-11184-f002:**
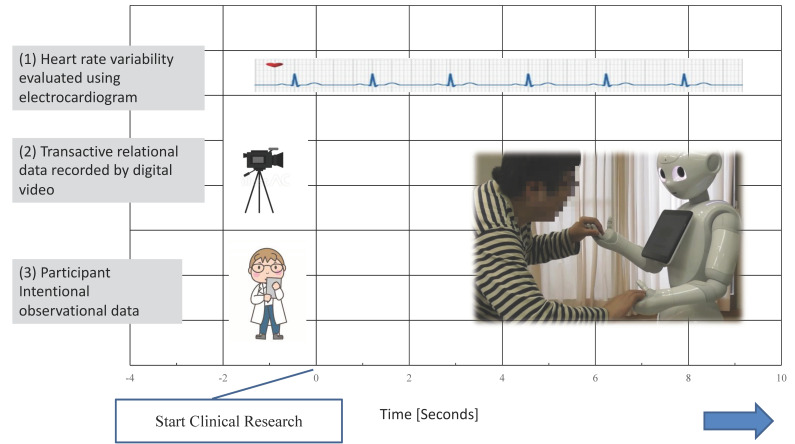
The framework of simultaneous research data collection using IOCRD.

**Figure 3 ijerph-18-11184-f003:**
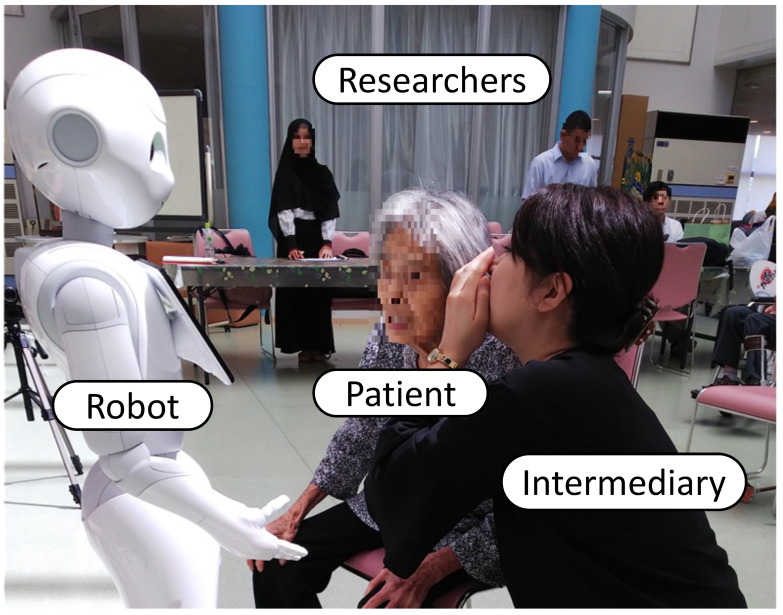
Transactive relations among intermediary, healthcare robots and older adults.

**Figure 4 ijerph-18-11184-f004:**
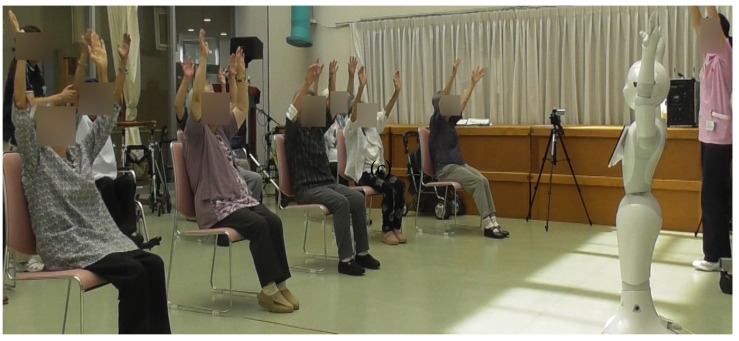
Scene of range-of-motion of the shoulder exercises instructed by Pepper robot.

**Figure 5 ijerph-18-11184-f005:**
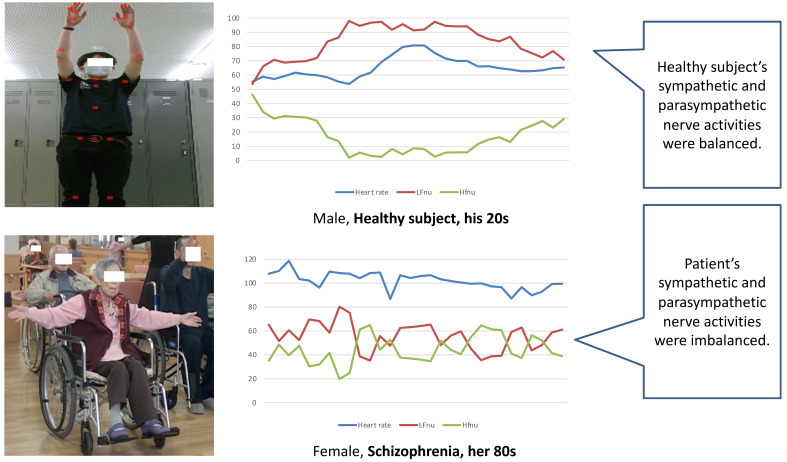
Synchronized image of (1) HRV evaluated using ECG, and (2) transactive relational data recorded by digital video recording during range-of-motion exercise. Note (Figure legend). The HRV analysis, acceleration score, and ROM of upper limbs were measured and analyzed using the GSM’s Bonaly Light. In healthy subjects, low-frequency (LF) normalized units (LFnu: sympathetic nerve activity) were increased and high-frequency (HFnu: parasympathetic nerve activity) was decreased during exercise. Patient’s antagonistic state of LFnu and HFnu was unclear. From this, it was considered that patients’ sympathetic and parasympathetic nerve activities were imbalanced by reduced HRV.

**Table 1 ijerph-18-11184-t001:** Differences between the IOCRD and other research designs.

No.	Items	Quantitative	Qualitative	Mixed Method	IOCRD
1.	Approach	Quantitative	Qualitative	Quantitative and qualitative	Quantitative, qualitative, and intentional observation with advanced technologies
2.	Aim	To predict and prescribe	To describe and explain	To obtain a broader perspective	To address the transactive engagement phenomena of human-intermediaries-robot and advanced technologies
3.	The unique feature	Measurement	Lived experience	Integration of quantitative and qualitative data	Simultaneous data collection and generation, and intentional observation using advanced technologies promoting innovation

## Data Availability

The data presented in this study are available on request from the corresponding author. The data are not publicly available due to privacy and ethical restrictions.

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
