# Peer review of "Intentional Observational Clinical Research Design: Innovative Design for Complex Clinical Research Using Advanced Technology"

_ijerph, 2021, doi:10.3390/ijerph182111184_

Round 1

Reviewer 1 Report

This paper aims to discuss the development and application of a new research design - the Intentional Observational Clinical Research Design (IOCRD). Data sources to develop the IOCRD were derived from surveyed literature on the past decade, focusing on clinical nursing research and relating robotics to nursing and healthcare practice.

In fact, the research topic is important for the setting where the study is carried out. In recent years, there has been a very significant advance in the research of nursing and health systems.

I understand that the research carried out by the authors brings out a need to create a standardized research design model. However, I do not consider that the proposal presented is valid following the following arguments:

1. IOCRD is a valid discussion proposal, but with a contribution to be verified.
2. IOCRD needs to be compared in practice with other designs.
3. Propose the IOCRD to a technical agency for study and eventually a standardized model. Ex: ISO or IEEE.
4. IOCRD can be used as a guideline model or as a basis for other research.
5. Propose the IOCRD to a discussion board for future use.

Author Response

Thank you very much for your valuable comments and suggestion. Please see our responses in the attachment.

Reviewer 2 Report

Dear authors, I was delighted to critically evaluate your manuscript. This manuscript discusses the development and application of the Intentional servational Clinical Research Design and shows the differences to Mixed Methods Design.

The manuscript is innovative and highly relevant for nursing research but also for other disciplines that develop and test new technologies for the health care sector in a multidisciplinary manner. Your design is not only important for use with robotics technology, but also for other new technologies where real-time data is measured. The manuscript is very well written and I have no suggestions for improvement.

Author Response

Thank you very much for your encouraging comments. Please see our response in the attachment.

Reviewer 3 Report

Thank you for requesting my participation in reviewing the manuscript « Intentional Observational Clinical Research Design: Innovative Design for Complex Clinical Research Using Advanced Technology .»

The article describes a clinical research design that answers complex research questions using advanced technology. Although difficult to read at times due to technical and special terms, the manuscript is adequately written. It would be of apparent interest to researchers in the field of advanced technologies in health care.

The article is primarily descriptive and is based on pre-existing theories and practical examples. It also emphasizes the advantages of IOCRD over other research models and its area of applicability.

It would be interesting to point out, in a short paragraph, the limitations of IOCRD (resources needed? Biais?).

I have no significant issues to highlight. The article is acceptable at the editor's discretion based primarily on the needs and demands of the journal.

Thank you

Author Response

Thank you very much for your valuable comments and suggestions. Please see our response in the attachment.

Round 2

Reviewer 1 Report

The authors provided answers.